# When Regular Education Is Not Effective and Conflicts Arise Between Systems: The Importance of Independent Educational Evaluations

**DOI:** 10.3390/bs15050594

**Published:** 2025-04-29

**Authors:** Dudley J. Wiest, Deven I. Landeros, Grahamm M. Wiest

**Affiliations:** 1Department of Special Education, Rehabilitation, and Counseling, California State University, San Bernardino, CA 92407, USA; 2Department of Child Development, California State University, San Bernardino, CA 92407, USA; deven.i.landeros@gmail.com; 3Miller Children’s & Women’s Hospital, Long Beach, CA 90806, USA; gwiest@memorialcare.org

**Keywords:** independent educational evaluation, ecological assessment, special education

## Abstract

Addressing the educational needs of children with learning challenges is often a complex issue, with few clear-cut accommodations that lead to high levels of interventional efficacy. This is especially true within the context of the child’s own developmental change, a dynamic school setting, and numerous family considerations. As a result, it is not uncommon for there to be disagreements among the school’s and the family’s perspective on how to best address the child’s educational, socio-emotional, and general developmental needs. This paper addresses this common occurrence in the United States public school system and discusses a model for how independent educational evaluations can be conducted to best inform the process that is intended to lead to appropriate and effective educational support for a student.

## 1. Introduction

The Individuals with Disabilities Education Act (IDEA), which was originally designated as Public Law 94–142 (Education for All Handicapped Children Act) in 1975, was authorized by the federal government (although education is constitutionally a “states” right) and subsequently implemented at the state level (with federal money to induce collaboration) to address the inadequacies of education for children who demonstrated mild to severe disabilities related to learning and education. The link between culture at large, the state and federal government, micro-systems (e.g., the home and the school), and the community were conceptualized as being interactive and were intended to support youths with disabilities from ages 3 to 21. Nonetheless, there can be systemic breakdowns which result in conflicts relative to eligibility, appropriate types of and intensity of interventions, and placement options for students identified with disabilities. As such, one of the salient rights for parents is the ability to request an independent education evaluation (IEE), funded by the district to settle such disputes. This article will present an ecologically based model for conceptualizing school-based concerns, delineate the essential legal features and sequencing of potential special education evaluation and placement, provide a neuropsychological approach for assessment, and describe a competence-based model for student interventions.

## 2. Background of Laws and Conflicts Surrounding Special Education

### 2.1. Early History of Special Education

Prior to the establishment of the Individuals with Disabilities Education Act in 1975, the right to an education granted by the states was not guaranteed for all. Children with disabilities could be excluded from attending school if they failed to meet certain standards, such as having a mental age of at least five years (determined by IQ testing) or having the ability to walk or use the toilet independently. Furthermore, children who passed these admissions standards yet struggled with behavioral issues (e.g., hyperactivity) were often expelled from school due to the burden their condition placed on their teachers and peers ([8]). Districts at the time had little means to effectively assist these children; thus, it was easier to exclude them completely than to make accommodations. While some wealthier families were able to enroll their children in private schools designed to educate those with disabilities, caregivers who could not afford to do so were left with two choices: either keep their child at home or institutionalize them ([8]).

After decades of controversy regarding the exclusion of children from participating in school due to their disabilities, two landmark cases helped pave the way for the development of the IDEA. The first case, [13] ([13]) took place when P.A.R.C. filed a lawsuit against the state of Pennsylvania on the behalf of parents whose mentally disabled children were blocked from participating in public education. The federal court ruled in favor of P.A.R.C., granting public school access to mentally disabled children and requiring the state to locate all those who were previously excluded (and still of school-age) to offer them a free education adapted to the child’s abilities. Additionally, the court’s decree mandated that parents be notified before their child is placed in a special education (SPED) program while also allowing for parents to voice their disagreements with such placements through an impartial hearing. The second case, [11] ([11]), set further precedents for educating children with disabilities. The court ordered that all children with physical, mental, or behavioral disabilities (regardless of severity) should be granted a free, public education program designed to address their unique needs via a proposed outline similar to an Individualized Education Plan (IEP). Together, the P.A.R.C. and Mills decisions opened the door to special education reform across the country, eventually pushing the United States Congress to protect the equitable education of children with disabilities ([8]).

### 2.2. The Establishment of the IDEA

In 1972, the United States Senate introduced the bill that would later become the Education for All Handicapped Children Act of 1975 (No. 94-142). Before the bill became law, it was estimated that out of the eight million disabled children in the United States, more than half were not receiving an appropriate education; further, another one million had been excluded from the public school system entirely ([5]; [8]). To help combat these staggering figures, four declarations were written into No. 94-142: (1) all children with disabilities would be offered a free and appropriate education and related services that are specially designed to accommodate their specific needs, (2) the federal government would assist states and local districts in providing special education to their citizens, (3) the rights of parents and students would be protected, and (4) special education efforts would be evaluated for their effectiveness. In 1990, the act saw several amendments, including changing the name to the Individuals with Disabilities Education Act and replacing the term “handicap” with “disability” throughout the document. The IDEA was officially signed into law in 1997, and additional amendments were added in 2004 and 2015 which expanded on recognizing the role that culture and linguistic diversity plays in designing and implementing special education programs ([8]). While there continues to be controversy regarding this critical legislation, it nonetheless sets forth basic principles regarding special education.

### 2.3. IDEA Core Principles

At its core, the IDEA has six foundational principles: appropriate evaluations, free and appropriate public education (FAPE), individualized education programs (IEPs), least restrictive environments (LREs), including parents and students in the decision-making process, and procedural due process ([14]). Since the IDEA’s purpose is to offer free special education to all children with disabilities, the process begins with locating and evaluating students to determine their eligibility for services, also known as the *child find* requirement ([8]). Each state must fulfill this responsibility which includes locating students who are homeless, in state custody (e.g., foster care, institutions), attending private school, transient, and those suspected of needing special education but are not currently identified ([8]). After a student is located, they are evaluated to determine whether (1) they have a disability which falls under the disability category list outlined by IDEA and (2) there is a need for special education services due to the disability ([8]). If a child is found to meet both qualifications and parents agree to have their child receive special education, establishing a FAPE for the student can begin.

FAPE is the key principle of IDEA because it establishes the fundamental tenets of the law itself; that is, it guarantees a free and appropriate public education for all disabled children between the ages of three and twenty-one. The “free” aspect of FAPE denotes that children who qualify for a SPED program are offered it at no cost to their family; instead, the educational experience is funded by a mix of federal, state, and local government agencies. Each step of the FAPE process is covered with public funds; these steps include the initial evaluation testing; the design, implementation, and updating of the IEP; and all related (necessary) services (e.g., speech and language therapy, occupational therapy, psychological services, physical therapy, etc.) that are required for the child to fully participate in their education ([14]). Next, the appropriateness of a special education program is determined by specifically addressing a student’s unique educational needs. This is accomplished by completing initial psychological and educational evaluations to establish the student’s baseline strengths and weaknesses. A group of educational professionals referred to as the Child Study Team (CST) then use the results of the evaluation to design an IEP that details the education plan for the student, realistic goals the CST expects the student to accomplish, and any accommodations the student will need to meet these goals. Additional educational requirements such as technology aids (i.e., cochlear device, laptop, etc.) and related services are also included if they are believed to be required for the student’s educational success. IEPs are reviewed annually and updated as needed based on continued observation and evaluation by the CST in order to adapt to the child’s physical, mental, and educational development ([14]).

Finally, the IDEA promises to supply funding for and assure the quality of special education programs that take place in the public school system. Two factors determine where a student will be placed: (1) proximity to the child’s home and (2) whether it offers the LRE for that particular child’s needs. Additionally, the local educational agency (LEA) may agree to place a student in a private school program only if an appropriate public option is not available. In these instances, the private program, along with any necessary services, are fully funded and must comply with all IDEA requirements. Regardless of the program a child attends, placements are reviewed annually to ensure goodness of fit and to insure a match to the developing child’s abilities ([8]). While the IDEA’s use of FAPE to place a child in the most appropriate special education program seems effective in theory, when it comes to the real-world application of the law, the system is not without its flaws.

### 2.4. Sources of Conflict

Although the IDEA and the implementation of FAPE were established to help educate children with disabilities in the most appropriate environment possible, several aspects of the act were designed in such a way that they have the potential to create conflict between parental beliefs and what the law dictates. The first major area of conflict begins with determining the initial eligibility of a student to receive special education. As was stated in [15] ([15]), there are no definitive criteria for determining whether a child is eligible for special education. While states are allowed to expand on their eligibility criteria, it can still lead to confusion as to which children are and are not able to receive services. Furthermore, issues also lie in the types of models used to decide a student’s eligibility. For example, when reviewing psychological and academic achievement tests, school psychologists will most often employ the discrepancy model which looks at the difference between a student’s IQ and academic achievement scores. If there is a 1.5 standard deviation or greater difference between the scores, the child is diagnosed with an LD and can begin the IEP process. Unfortunately, because the discrepancy model requires there to be a preexisting gap in what the child should be capable of achieving and what their true scores are, this means that the child will have already shown signs of failing long before they are formally recognized and begin receiving services, lending to this model often being referred to a “wait to fail” approach. Another model to qualify students for special education is known as response to intervention (RTI). This model is sometimes employed for children who have shown obvious signs of struggling in school and is used to combat the likelihood of them failing before an official evaluation can be provided ([14]). While this model is implemented with the hope that the child will respond positively to the intervention and thus not need further services, the interventions are often unsuccessful because they are not specially curated to the student’s unique needs as an IEP would be, causing the child to fail nonetheless. This lack of a quick, efficient, and fully agreed upon method of evaluating struggling students can be one source of major conflict between parents and the district. That is, when there is no “black-and-white” means (or standards) of making a diagnosis regarding a child’s educational needs, there is space for alternative conclusions (and thus conflict).

Another source of contention can arise because of where special education is provided; that is, where is a student placed after being identified? As outlined by the IDEA, children with special education services are to be placed in the LRE possible for them to be able to achieve their IEP goals. Broadly speaking, there are four types of settings in which special education takes place: general education classrooms with push-in (most inclusive), resource rooms (pull-out for specialized support), self-contained classrooms (in which a child may share recreational time with non-disabled peers but is educated separately), and alternative placements (a separate environment from non-disabled peers that is least inclusive) ([2]). When deciding on where to place a student, the CST must take into consideration the extent to which the child will be able to effectively achieve their IEP goals as well as the impact the child will have on the classroom environment ([2]). For example, a student with extreme behaviors (as seen in some cases of autism spectrum disorder) is likely to have a difficult time adjusting to the curriculum of a general education classroom and would likely impact (negatively) other children’s learning experiences in that classroom. In such cases, a self-contained classroom placement is likely to most effectively support the child’s attainment of IEP goals. Although this may be what the CST and the district believe is best for the student’s overall academic success, parents may disagree, pushing for their child to be placed in a general education classroom for the sake of inclusion. Alternatively, in the previous example, the parents may wish their child to attend a private school that specializes in intervention practices targeted for a specific exceptionality; however, the school district considers itself to be fully capable of meeting the student’s educational needs. In either of these examples, placement considerations may become a significant source of conflict between the school and parents.

The last major source of conflict between parents and the district concerns the types and intensity of interventions that the CST includes in a student’s IEP. Students are given an IEP with realistic yet challenging goals for the year based on their initial eligibility testing. The special education program, the accompanying services, and the intensity of both are designed to help the student achieve these goals within the timeframe based on best practices and research-informed interventions. Each year, these goals and services are reviewed and may be increased, reduced, or remain the same depending on the student’s progress. In the past, some schools would underestimate the needs of a student, inappropriately advancing them when they were not ready or not supplying the appropriate services or intervention intensity the child required ([9]). This practice was not outside the rules of the IDEA because, as long as the school could prove they were supplying a FAPE, even minimal progress was considered sufficient. Thanks to [6] ([6]), this use of *de minimis* is no longer acceptable; however, there is continued dispute over whether IEP goals are challenging enough and if the progress that a child makes is the result of their special education or simply them finding ways to get by in their academics. Unless parents notice their child is struggling under their current IEP and present their concerns to the CST, it is likely the child will fall further behind since they lack the proper support. Unfortunately, many families do not speak up due to feeling that their opinions are not valued compared to those of the “expert” members of the CST ([7]). However, when parents do advocate for an appropriate FAPE, their child’s unresolved educational challenges may be better addressed. One means of advocating on a child’s behalf is via an independent educational evaluation (IEE).

### 2.5. Introduction to Independent Education Evaluations

The complexity of the IDEA and the lack of congruence with its application (across educational settings) can understandably be frustrating for parents who desperately wish to see their struggling child succeed. It is this discontentment with the system which leads some parents to request an IEE in the hopes of getting their child the appropriate educational support they require. An IEE is conducted by an independent third-party licensed or credentialed professional with expertise in the field most closely related to the contested issue (e.g., a neuropsychologist sought for a case regarding special education eligibility). Typically, an IEE occurs because (1) the district agrees with the parent’s request and hires the IEE assessor themselves or (2) the school district disagrees with the need for an IEE and the parents opt to pay for the evaluation on their own. Once a clinician has been hired, they will review all previous decisions regarding the student’s education (including IEPs), assessments, and behavior records. Using this information, the clinician will design a specific battery of testing and observations in order to document the student’s educational needs to the greatest extent possible. Once this is complete, the clinician will compile their findings, use them to determine whether the district appropriately applied best practices for the remediation of the potential deficit, and formally offer their suggestions for how the district should respond to the results of the IEE. In general, IEEs are lengthy and involve extensive testing using norm-referenced measures, interviews, and observations in all relevant settings. It is via this process (that is not limited by school district resources) that a full and complete student educational profile is established.

## 3. Utilizing a Systems Approach When Conducting an IEE

Since IEEs are conducted with the intention of appropriately assessing a child’s educationally related strengths and weaknesses, it is necessary to understand not only the child but how their environment and the people around them have influenced their development throughout their lives. The first author, a licensed and board-certified school psychologist who specializes in IEEs, utilizes Bronfenbrenner’s Ecological Systems Theory ([4]) (see Figure 1) to structure the assessment with the objective of fully understanding the multidirectional influences of intrapersonal, interpersonal, and contextual factors impacting a child’s current school experience. In identifying and fully accounting for these relationships, the IEE is intended to affect the development of an IEP.

The most widely used visualization of Bronfenbrenner’s systems model consists of six interrelated circles each containing different relational and environmental factors that influence how a child develops (and that the child impacts, in turn). The further a circle is away from the child, the more indirect the influence. The initial level of understanding (within an ecological approach) encompasses the child’s individual characteristics (e.g., their physical, cognitive, language, and intellectual abilities; their self-perceptions; and their social–emotional development). From there, we delve into the microsystem, which includes the child’s interpersonal relationships (e.g., with their parents, siblings, peers, and teachers). The mesosystem appears next and represents the connections among the agents that comprise the microsystem; within this system, an ecological perspective considers how each of the agents interact with one another, as well as how they are affected by indirect entities (e.g., mass media, parent’s workplace, community welfare) of the exosystem (that are a part of this system but do not likely have a direct impact on the child, nor does the child directly influence them). Within the macrosystem are the influences that are most indirectly related to the child yet still hold a notable influence on the child’s development. Examples of macrosystem components may be the general attitudes and ideologies regarding education in the community where the child resides, the current cultural and/or political zeitgeist, and overall economic climate in which the child lives. The last level of the model, the chronosystem, considers the movement of time, both for the developing person as well as the people and environment they are surrounded by. Each of the levels (or systems) interacts with the others to reveal a holistic view of the child.

To help conceptualize a systems approach, imagine there is a child who has a specific learning disorder. They have supportive parents and attend a school with a reputable SPED program (microsystem). The parents attend every IEP meeting and work with the school to find the best accommodations for their child (mesosystem). Fortunately, the local government has allotted funds for the school to purchase new supplies for SPED students (macrosystem) and has guaranteed these funds for the next 10 years (chronosystem). Under these circumstances, this child is well supported and is likely to be successful in school. However, if the evaluation is not comprehensive or the IEP is not well suited to the child’s developmental and educational needs (or many other potential concerns), the outcome for the child may be very different. This hypothetical example is often repeated in many real-life experiences of K12 students and their families. In utilizing an ecological approach, the aim is to have an assessment process that results in appropriate evidence-based interventions that best meet the needs of a student. It is for this reason that professionals contracted to perform IEEs need to understand on a fundamental level that there is no “one-size-fits-all” when it comes to assessing children for special education services.

Although the ecological model has been used in many areas of research, Bronfenbrenner himself recognized the benefits of utilizing a systems approach in education ([3]). Bronfenbrenner challenged the idea of implementing research in schools that had only been tested in a controlled laboratory environment and thus worked to promote a system that could reveal the complex layers of how students learn within their own lived experiences and physical settings. He understood that the dynamics between a student and the contexts of their school environment (i.e., teachers, peers, institution quality, parental support) influence each individual in a unique way and that it is vital that this holistic view is considered when making best-practice decisions for a child’s academic path. Nearly fifty years later, Bronfenbrenner’s theories on education are still relevant, specifically in the context of how we can think systematically and holistically while balancing mechanistic models of pathology (i.e., using diagnostic criteria) when performing IEEs for children with disabilities. Although the concept of checking off criteria to determine one’s eligibility for special education is a more straightforward and seemingly “equal” method, the complexity of human development and the influence of one’s environment often renders this method unreliable. Utilizing these unreliable methods can lead to inaccurate results and thus inappropriate diagnoses, placements, goals, and interventions for children who desperately require appropriate accommodations to succeed.

## 4. School Neuropsychological Independent Educational Evaluation

Given the premise for most IEEs (i.e., conflict/disagreement between the school and a student’s parents regarding the child’s educational needs), it is essential that this type of evaluation is comprehensive and empirically grounded. In other words, it is essential that IEEs have a solid clinical basis and are empirically (data-) driven. The figure below (Figure 2) is based upon a neuropsychological perspective of academic achievement. This model was developed by the first author and is based upon [10] ([10]) conceptualization of neuropsychological assessments.

We conceptualize school neuropsychological assessment from a perspective that emphasizes human development and the interactions between the genetics of a person, the environment, and the biological substrates than emanate from this relationship. This model integrates traditional ideas of neuropsychology with an emphasis on measuring brain function and processes utilizing psychometrically strong indicators (e.g., intelligence tests, executive function indices, visual-motor tests, etc.). For a review of neuropsychological assessment, see [1] ([1]), [10] ([10]), and [12] ([12]). Within this assessment model, it is assumed that predispositions toward temperament and intelligence/cognitive factors (e.g., linguistic, visual/spatial, memory, speed, abstraction processes) mature and progress with stable environmental support such as safe communities, quality schools, financial resources, a stable family environment, and consistent, supportive feedback from adults who scaffold and foster the developmental experiences of the child. Moreover, optimal development is assumed to be supported by stability of physical health, nutrition, prenatal care, and safety from brain injuries. Thus, a school neuropsychological evaluation begins with an extensive interview with the parents to (as fully as possible) ascertain this developmental history; frequently, additional interviews may occur during the course of the evaluation. Assessment at the next level includes determining the efficacy of attentional processes (including selective attention, sustained attention, attention shifting, and attentional capacity) as well as the sensory systems associated with visual, auditory, gustation, olfactory, and sensation processes. We conceptualize working memory processes as not only represented in executive skills but simultaneously interacting with numerous other cognitive processes such as memory (immediate and long-term, visual and auditory), language, visual/spatial awareness, executive refinement (planning, organization, abstraction, reason, emotional control, inhibition), and processing speed (see Figure 2). Thus, at this level of assessment, specific testing is completed for each of these abilities with the student. In general, testing is extensive and may involve the use full measures and/or specific subtests from multiple measures.

Ultimately, the IEE concludes with a review of data and an opinion based upon the referral questions, law, and data (gathered via interviews, testing, and/or observations). The conclusion is presented to both parties with recommendations.

It is essential to be mindful that an IEE is predicated on addressing conflict between the school district agency and the child/family. Thus, the examiner(s) must determine the proper interpretation of clinical disagreements which can include issues regarding the determination of eligibility for special education services, appropriateness of goals and intensity of service, types of intervention strategies, and placement. Clarity on these disagreements leads to a set of referral questions that guide the process of evaluation. These referral questions may include the following:What are the child’s intangible strengths and weaknesses?What are the child’s cognitive strengths and weaknesses within the context of the neuropsychological model presented above (see Figure 2)?What are the academic baseline skills in all areas?Are there mental health concerns that affect education?Where does learning break down?Does the potential breakdown have a negative impact on the child’s educational experience, thus necessitating SPED or Section 504 accommodations?Did the district appropriately evaluate, intervene, and apply best practices for the remediation of potential deficits?

## 5. Intervention

As part of the IEE process, evidence-based interventions are recommended to address the presenting concerns. These recommendations may be offered because the child is found to be eligible for special education services through the IDEA. Alternatively, the student may qualify for accommodation services via Section 504. Finally, some students may not be eligible to formally receive services but do, if fact, require some type of educational, behavioral, and/or mental health supports and treatments. We conceptualize the decision-making process for symptom amelioration as including necessary changes in the ecology of the child. For example, there may be parent education, attempts to more effectively integrate the home and school systems, or adding a skill for the child (such as music or sports) that highlights a strength in the child. Moreover, there may be recommendations specific interventions for problems such as attention (e.g., behavioral training of parents and staff, medication, or social skills training), reading (e.g., simultaneous multisensory approaches for reading development), social skills training (e.g., intensive outpatient treatment for coping skills and therapy), and executive skills training (e.g., tutors, computerized cognitive training programs, and teacher scaffolding of organization/planning). There is a wealth of resources regarding specific interventions; thus, it is vital for the clinician conducting the IEE to be well versed in the systems approach discussed previously.

Ideally, the interventions should follow our model of competence development and intrinsic motivation (see Figure 3). In this model, it is paramount that an individual develops a sense of mastery on tasks which are optimally challenging (i.e., tasks that are neither too difficult nor too easy). As a result of engaging in such tasks, the child experiences satisfaction from an evolving internal sense of self (that becomes stronger); further, they will likely receive positive feedback from teachers, parents, coaches, and peers about their performance. If there is a consistent cycle of successful mastery attempts, the child’s self is increasingly constructed into positive and internal precepts of competence, confidence, and stability that underlie the development of an independent and autonomous person. Conversely, when children are exposed to frustration, failure, and negative feedback from others and when there is a lack of mastery in social skills, academics, emotional regulation, and executive skills, their optimal development of a positive self is ruptured (with enough failure experiences). Thus, evidence-based interventions can and should be implemented at the points of breakdown (i.e., the “Wall of Demand”; see Figure 3). Moving the child to the other side of the “Wall of Demand” where the Competency Spiral (see Figure 3) can occur is crucial for personality formation and subsequent positive experiences in culture. The systems-based IEE described in this paper becomes an important foundation for new ways of understanding mastery and competence for the child. Moreover, the IEE process effectively engages all systems (e.g., the school and the family) to work together on interventions that support positive educational, intrapersonal, and interpersonal development for the child.

## 6. Illustrative Examples of IEEs

Two illustrative examples are presented here to demonstrate the systems-based IEE process described in this paper. Each of the examples highlight how the IEE provided further clarity on each child’s developmental and educational needs, which led to more effective support for each student. These examples are intended to call attention to the complexity of addressing a student’s educational needs and the importance of a holistic approach in meeting academic accommodations.

### 6.1. Jillian

A local public school district contacted our office to conduct an IEE for an 11-year-old fifth-grade child attending public school. After the contract was submitted and approved, the parent and district representatives were interviewed about perceptions of the problem. Jillian had a long-term diagnosis of autism spectrum disorder (ASD) which was first noted when she was a toddler. Subsequent early intervention procedures (both publicly and privately funded) such as occupational therapy, speech/language therapy, applied behavior analysis (ABA), social skills training, parent training, and preschool support were implemented. Jillian responded exceptionally well to these interventions and attended a parochial school through third grade, eventually transferring to public school in fourth grade. In the public school, the parents contended that Jillian required special education services to benefit from school. A school evaluation was conducted, and the school team noted little to no evidence that the early ASD was affecting school. Her intelligence was well above average, as were all academic skills. All cognitive measures of processing and social skills were above average, although she had some inconsistent attention. Jillian was an “A” student with above-average scores in state testing. Homework was consistently returned. Moreover, Jillian had a core group of friends and did not demonstrate unusual behaviors or mental health problems at school. The parents reported ‘meltdowns’ at home, primarily regarding homework. Jillian was active in sports, had supportive parents, and a younger brother who was also diagnosed with ASD, although more severe in presentation. The IEE evaluation concurred with most findings of the school. Jillian did have above-average to superior cognitive processing skills, intelligence, and academics except for attention. Ultimately, the diagnostic conclusion of the IEE was that Jillian met the criteria for attention deficit hyperactivity disorder (ADHD) and autism spectrum disorder (ASD), and there was evidence of a mild anxiety disorder as well. These diagnostic areas did not impede school efficacy at the time of the evaluation. The IEE supported a Section 504 plan with assistance with planning and organization as needed. The parents and school were encouraged to work collaboratively with Jillian on the types of homework she received, reinforcement systems for homework completion, organization, and intermittent counseling with the school psychologist as needed for anxiety and distress. The parents were provided books and referrals for parenting children with special needs and utilizing a behavioral family therapist to develop better skills (as a family) for tasks such as organization and transition. Additionally, the family was recommended to have a medication consultation for the subtle inattentive ADHD. Finally, the school and home were advised that when a transition to middle school was imminent, there be collaboration among the current school, the staff at middle school, and the parents to determine if additional intervention was required.

### 6.2. Andrew

Andrew’s parents were given permission to proceed with an IEE (initially parent-funded and subsequently reimbursed by the district) to determine if he required an alternative placement from his local public school. The parents contended that the speech and language services as well as specialized academic instruction (SAI) provided by the local school were not adequate to address his needs in school. They requested a more restrictive, nonpublic school setting specializing in treatment of neurodivergent youngsters with ADHD and ASD for their son. The historical records noted long-term problems since infancy with emotional dysregulation and tantrums which could last 90 min every day at home. After being diagnosed with ASD, he was provided special classes, ABA at home and school, speech and language interventions, and occupational therapy. Additionally, he was seeing a child psychiatrist and had daily medication. Although his skills improved in his elementary school, he demonstrated poor social skills. His academic skills were average in all areas except for written language, which required SAI pullout teaching; however, his skills were too advanced for a generic special day class. The parents noted that their son had become very depressed the previous year, had few friends, and experienced teasing by other students. Additionally, Andrew developed excoriation and picking at his face until his scabs openly bled during class. The IEE noted above-average intelligence with multiple weaknesses in cognitive processing, including attention, inhibition, working memory, sensory and fine motor control, and visual memory. Ultimately, Andrew was diagnosed with ASD, ADHD, a major depressive episode, and disruptive mood dysregulation disorder (DMDD). The IEE concluded that his goals and objectives were not met in the district setting, and after observing other placements in the district as well as the nonpublic school specializing in behavioral, social skill, and mental health treatment of neurodiverse children, placement in this more restrictive setting was recommended. The IEP team agreed; thus, Andrew changed schools and continues in that setting.

### 6.3. The Importance of Utilizing an Integrative Approach

In the illustrative examples above, the application of systems theory, the neuropsychological framework, and the intrinsic motivation model were utilized to better conceptualize a more complete illustration of each student’s past and present life circumstances, cognitive aptitude, and responsiveness to previous interventions to determine whether the IEP team took appropriate measures in addressing their unique needs. In Jillian’s case, it was critical to look beyond her satisfactory grades to understand the cause of her meltdowns at home. Although she had responded positively to prior intervention and her ASD and ADHD diagnoses were shown to not be impeding her school success, the information gathered from her parents on the anxiety Jillian was experiencing over homework was of notable concern, especially since she would be facing even greater challenges in the near future when transferring to middle school. Utilizing a systems approach and devising a plan with Jillian’s family to address these issues before she transitions allows her a greater chance of developing a sense of competence related to school and is likely to decrease her reliance on interventions in the future as a result of the strong collaboration between the school system and family.

As for Andrew, his past was riddled with multiple diagnoses and ever-changing interventions. While his grades were mostly average, he continued to struggle socially and emotionally, which in turn impacted his school success. By employing the neurological assessment model, the evaluator was able to determine that Andrew’s difficulties appeared to stem from his diagnosis (i.e., ASD, ADHD, DDMD, major depressive episode), the effects of which were impacting his cognitive functioning. After carefully considering his school experiences and settings, it was determined that a more restrictive and specialized environment with children similar to himself would likely best support Andrew’s educational and developmental needs.

## 7. Conclusions

In the twentieth century, Western culture began to prioritize the need for school services for all children, including those who had been previously excluded due to physical, mental, or emotional disabilities. As a result, laws such as PL 94-142 (now IDEA) and the Rehabilitation Act of 1973, Section 504, were instituted. Today, American culture continues to prioritize the rights of these individuals to receive a free appropriate public education and, much like the education system at large, it is no secret that this effort is fraught with the struggle of determining boundaries of the laws and the apportionment of monies for support programs. IEEs have been built into special education law to act as a form of “checks and balances” by mediating conflicts between the school districts who determine the eligibility for and provide the interventions for students with disabilities and the parents of said children when they believe their needs are not being adequately met. Although there are a variety of methods employed by evaluators when conducting IEEs based on their professional expertise and philosophical beliefs, there is one critical consideration that we should be consistently mindful of: conceptualizing a student’s developmental and educational trajectories and how they will respond to educational accommodations is a complex and uniquely individualized issue. Given this perspective, the utilization of a multi-perspective, multi-information approach that taps into relevant theoretical and conceptual perspectives, alongside a comprehensive neuropsychological assessment, can guide evaluators in fully understanding the sources of a student’s school-related challenges. Ultimately, this means that the school and the family can effectively untangle the myriad of complex relations that determine school success and establish a path to greater school success and individual growth.

## Figures and Tables

**Figure 1 behavsci-15-00594-f001:**
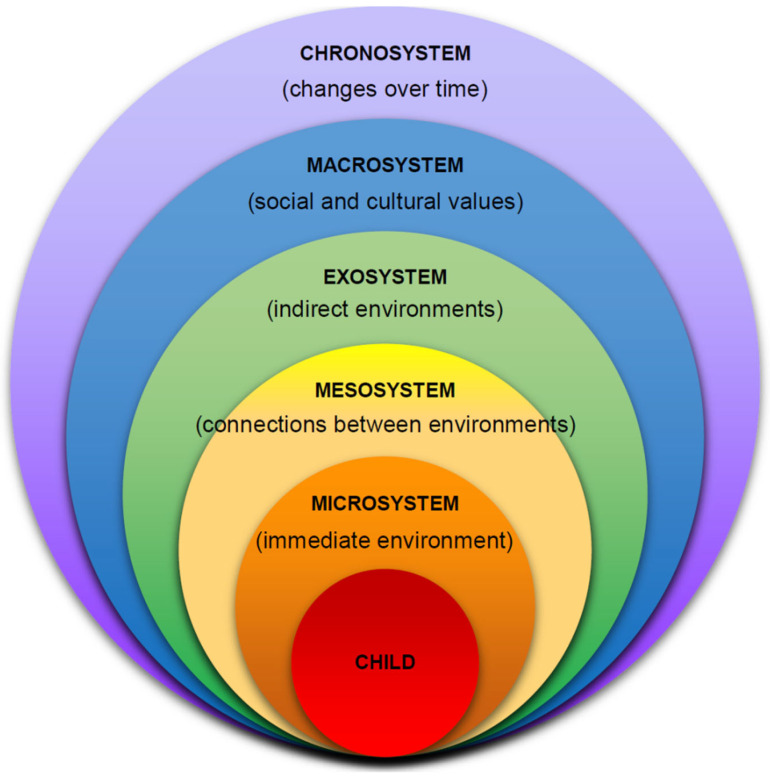
Bronfenbrenner’s ecological model.

**Figure 2 behavsci-15-00594-f002:**
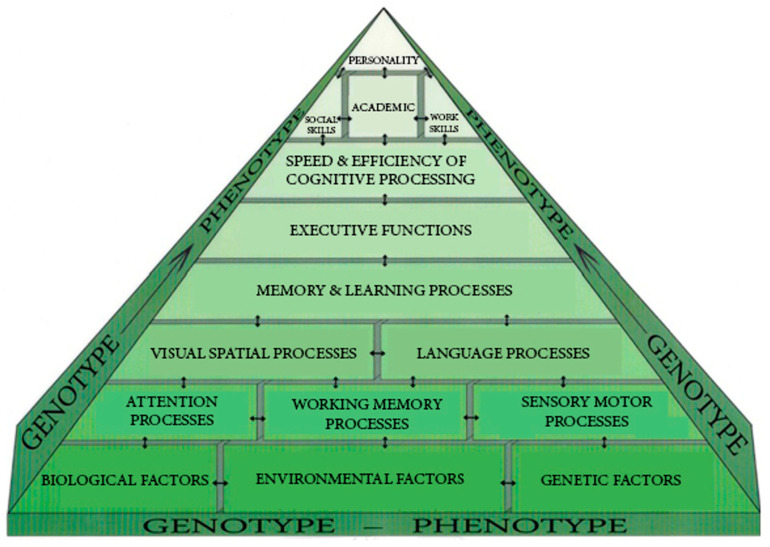
A model for a neuropsychological assessment of learning and achievement.

**Figure 3 behavsci-15-00594-f003:**
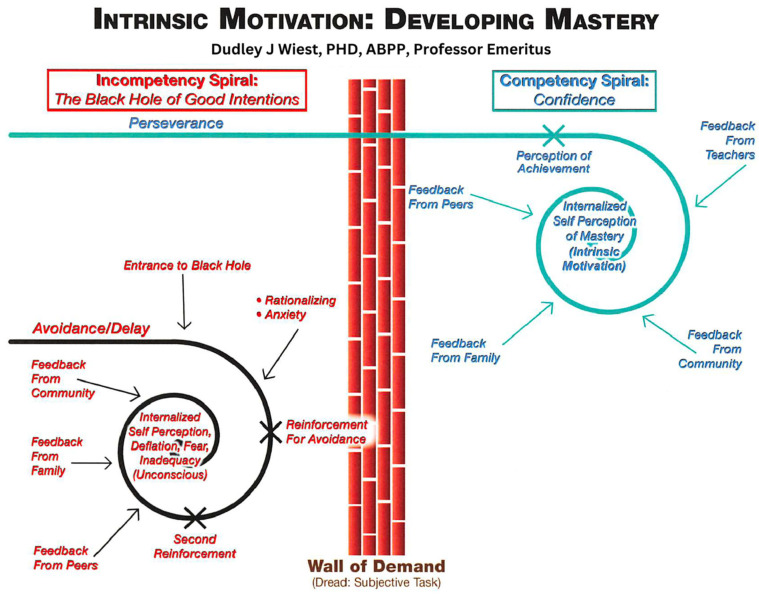
A competence-based model for mastery motivation.

## Data Availability

The original contributions presented in this study are included in the article. Further inquiries can be directed to the corresponding author.

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
