# Peer review of "When Regular Education Is Not Effective and Conflicts Arise Between Systems: The Importance of Independent Educational Evaluations"

_behavsci, 2025, doi:10.3390/bs15050594_

Round 1
Reviewer 1 Report
Comments and Suggestions for Authors
The paper presents a study of two cases, where it would be pertinent to include more specific details related to the types of tests employed, the time allocated for evaluation, and the intervention strategies used. These elements are crucial for strengthening the understanding of the methodological design, even if it is qualitative in nature and based on a case study approach.
The discussion on current legislation and the modifications implemented over time, as well as the previous and current theories underpinning the research, stands out. However, although the authors include references to legislation and theories, they do not present a strong argument to substantiate these connections convincingly.
Regarding the conclusions, it is recommended to review them to ensure they reinforce and justify the choice of theories presented and are clearly linked to the cases analyzed. This would help consolidate the internal coherence of the article and strengthen the relationship between the data presented and the interpretations made.
Author Response
We wish to thank the peer reviewers for their thoughtful and insightful comments on our manuscript. This manuscript is not an empirical project; it is a conceptual/theoretical paper that addresses a complex academic issue for students with learning differences. Within this student population, conflicts often arise between the school system and the family. As a result, Independent Educational Evaluations (IEE) are necessary to untangle incongruent views as to the best practices (i.e., evidence-based interventions) that should be considered for a specific student. The manuscript discusses a model for conducting IEEs that is solidly grounded in theory and neuropsychological assessment/evaluation principles. Additionally, two illustrative examples are provided to give context for the IEE approach that is discussed.
Because of potential misinterpretation of this manuscript reporting on a completed study, the wording “case studies” have been replaced with “illustrative examples” where relevant. Moreover, we have added a new section (i.e., section 6.3, The Importance of Utilizing an Integrative Approach) to further delineate a wholistic approach to addressing complex academic issues.
Finally, the conclusion has been revised to reinforce the key points of the paper and highlight the primary focus of this manuscript – a systems- and theoretically-based IEE approach is an effective tool for resolving the complex conflict that arises between parents and the school when a student’s educational needs have not been fully addressed.
Responses to specific review comments are below:
- Addressed above in paragraphs 1 & 2.
- No change was made to the manuscript regarding this comment because the legislation and theories are not intended to be "connected". The content about legislation is intended to highlight the rights of families with regard to public education. As such, it illustrates how conflict can arise between a family and a school district. The theory content (i.e., ecological approach and neuropsychological approach) reflects how the authors approach conducting an IEE.
- Addressed above in paragraph 3.
Thank you for your consideration of this manuscript. We believe that it represents a significant and timely contribution that is of relevance to practitioners, researchers, and families. We look forward to your decision.
Reviewer 2 Report
Comments and Suggestions for Authors
I have reviewed your paper and believe that with some revisions, it could be made even stronger and more suitable for publication. These changes will help clarify your key points and enhance the overall quality of the work.
1. My suggestion to the authors of the paper is to provide a more in-depth analysis and comparison between the two case studies of Jillian and Andrew, particularly explaining why, despite both having similar diagnoses (Autism and ADHD), their educational needs and outcomes were vastly different. This analysis could include the following factors:
A. Cognitive Differences: Exploring how Jillian’s higher intelligence and cognitive skills contributed to her success in a public school setting, while Andrew’s attention deficits and emotional challenges required a more specialized educational environment.
B. Previous Interventions: Detailing whether Andrew’s school made any modifications to his educational plan before recommending an alternative placement, and what strategies were attempted.
C. Differing Evaluation Methods: Explaining how the Independent Educational Evaluation (IEE) might have identified issues that previous school assessments did not capture.
Adding this comparative discussion would help readers not only understand the differences between the two cases but also the reasons behind those differences. It would enrich the discussion and further emphasize the importance of independent evaluations in identifying individual needs and providing appropriate support.
2. The paper mentions that Andrew engaged in self-harm in class but does not provide details about the school’s response to this behavior prior to the IEE. It would be helpful to know if the school made any efforts to address his mental health concerns or implemented interventions before recommending a change in placement.
3. The paper would benefit from a clearer discussion of practical applications, particularly focusing on how educators, parents, and policymakers can use the findings from the case studies. It is suggested that a dedicated section be added to explore proactive measures, such as whether schools can implement some of the IEE recommendations before parents resort to an independent evaluation. Furthermore, it would be valuable to discuss early warning signs that can help prevent students from reaching a point where an IEE becomes necessary. These additions would significantly enhance the paper’s relevance and applicability, offering actionable insights for those supporting neurodiverse students. By incorporating these suggestions, the paper would provide not only theoretical insights but also practical guidance for real-world implementation.
4. Since the paper heavily relies on two case studies, it would be beneficial to address whether these cases are representative of broader trends in special education. It is important to acknowledge the limitations of drawing conclusions from a small sample size. Additionally, the authors could clarify if any generalizations are being made and if so, whether they should be interpreted with caution.
5. The paper mentions models of human development, systems approaches, and neuropsychological assessments, but these frameworks are not clearly connected to the case study analysis. To strengthen the paper, consider including a brief section that explicitly links these theories to the findings from Jillian’s and Andrew’s evaluations.
6. The conclusion would be stronger with explicit statements summarizing key findings, such as the value of Independent Educational Evaluations (IEEs) in identifying overlooked disabilities, and the importance of early intervention and collaboration between parents and schools. These points would clarify the practical implications and strengthen the paper's impact.
Thank you!
Author Response
We wish to thank the peer reviewers for their thoughtful and insightful comments on our manuscript. This manuscript is not an empirical project; it is a conceptual/theoretical paper that addresses a complex academic issue for students with learning differences. Within this student population, conflicts often arise between the school system and the family. As a result, Independent Educational Evaluations (IEE) are necessary to untangle incongruent views as to the best practices (i.e., evidence-based interventions) that should be considered for a specific student. The manuscript discusses a model for conducting IEEs that is solidly grounded in theory and neuropsychological assessment/evaluation principles. Additionally, two illustrative examples are provided to give context for the IEE approach that is discussed.
Because of potential misinterpretation of this manuscript reporting on a completed study, the wording “case studies” have been replaced with “illustrative examples” where relevant. Moreover, we have added a new section (i.e., section 6.3, The Importance of Utilizing an Integrative Approach) to further delineate a wholistic approach to addressing complex academic issues.
Finally, the conclusion has been revised to reinforce the key points of the paper and highlight the primary focus of this manuscript – a systems- and theoretically-based IEE approach is an effective tool for resolving the complex conflict that arises between parents and the school when a student’s educational needs have not been fully addressed.
Responses to specific review comments are below:
- Addressed above in paragraph 1.
- The authors believe that additional details about how the school addressed Andrew’s mental health concerns are not relevant to the main purpose of the illustrative example, which is to demonstrate how an IEE can be conducted using the methods described in the paper. Thus, no changes were made regarding this comment.
- No change was made to the paper for this comment because this manuscript is not focused on practical implications. Its focus is in reference to legislation related to student’s rights, conflicts between families and schools regarding a student’s educational experience, and the role of IEEs when these conflicts arise.
- Addressed above in paragraphs 1 & 2.
- This comment was addressed in the manuscript through the addition of section 6.3 The Importance of Utilizing an Integrative Approach.
- Addressed above in paragraph 3.
Thank you for your consideration of this manuscript. We believe that it represents a significant and timely contribution that is of relevance to practitioners, researchers, and families. We look forward to your editorial decision.
Round 2
Reviewer 1 Report
Comments and Suggestions for Authors
The authors have included the references in the text, in an effort to clarify and update the subject on which they are reflecting.
Some paragraphs have been expanded, especially in the introduction and conclusion.
Only two cases (examples) are presented, making it a theory-based article.
It would be interesting to know the objectives of the research.
Author Response
We thank the reviewers for their careful consideration of our manuscript and their thoughtful comments. Below, please find our response to reviewer’s thoughts.
As noted by the reviewer, this manuscript is a theory-based article. Therefore, there are no research objectives discussed (as it is not relevant). This paper describes a model for how independent educational evaluations can be conducted to address a student’s educational needs and academic accommodations.
Reviewer 2 Report
Comments and Suggestions for Authors
Thank you for addressing the comments and refining the discussion. I appreciate your efforts in strengthening the clarity and depth of the analysis. The contrast between Jillian's and Andrew's cases raises an important question: Do integrative approaches in education prioritize individualized accommodations over broader systemic reforms?
Thank you
Author Response
We thank the reviewers for their careful consideration of our manuscript and their thoughtful comments. Below, please find our response to reviewer’s thoughts.
The focus of an individual educational evaluation is the student his/herself (themselves). Thus, this manuscript does not address broader systemic reforms. However, this does not reduce the importance of systemic reforms; it is simply not the focus of this manuscript.
Again, thank you for your consideration of our manuscript. We believe that our theory-driven discussion of independent educational evaluations is an excellent match of the special issue. We look forward to your publication decision.